

# Enhancing the prediction of vitamin D deficiency levels using an integrated approach of deep learning and evolutionary computing

Ahmed Alzahrani[1] and Muhammad Zubair Asghar[2]

[1] Department of Computer Science, Faculty of Computing and Information Technology, King Abdulaziz University, Jeddah, Saudi Arabia
[2] Gomal Research Institute of Computing (GRIC), Faculty of Computing, Gomal University, Dera Ismail Khan, KP, Pakistan

## ABSTRACT

Vitamin D deficiency (VDD) has emerged as a serious global health concern that can lead to far-reaching consequences, including skeletal issues and long-term illness. Classical diagnostic approaches, although effective, often include invasive techniques and lacks to leverage the massive amount of healthcare data. There is an increasing demand for noninvasive prediction approaches for determining the severity of VDD. This work proposes a novel approach to detect VDD levels by combining deep learning techniques with evolutionary computing (EC). Specifically, we employ a hybrid deep learning model that includes convolutional neural networks (CNN) and bidirectional long short-term memory (BiLSTM) networks to predict VDD data effectively. To improve the models effectiveness and guarantee the optimal choice of the features and hyper-parameters, we incorporate evolutionary computing methods, particularly genetic algorithms (GA). The proposed method has been proven effective through a comprehensive assessment on a benchmark dataset, with 97% accuracy, 96% precision, 97% recall, and 96% F1-score. Our approach yielded improved performance, when compared to earlier methods. This research not only push forward predictive healthcare models but also shows the potential of merging deep learning with evolutionary computing to address intricate health-care issues.

## INTRODUCTION

Vitamin D deficiency (VDD) is a significant global health issue, associated with conditions ranging from skeletal disorders to chronic illnesses. Traditional diagnostic methods for VDD rely heavily on invasive techniques like blood tests, which, while effective, are often costly, uncomfortable, and inaccessible for many populations. Recent advancements in non-invasive predictive healthcare solutions, driven by machine learning and artificial intelligence, provide a transformative opportunity to address these limitations. These techniques enable the analysis of vast healthcare datasets to offer accurate, scalable, and patient-friendly diagnostic tools. Therefore, it is essential to develop more efficient non-

Corresponding author
Muhammad Zubair Asghar,
mzubairgu@gmail.com

invasive predictive systems that can utilize the extensive health data collected in both clinical and non-clinical settings to predict vitamin D deficiency levels accurately (*Sancar & Tabrizi, 2023*).

Recent progress, in the field of deep learning shows potential for improving the forecasting and comprehension of intricate health issues such as vitamin D insufficiency. Advanced machine learning (ML) machine learning techniques, such, as convolutional neural networks (CNN) and bidirectional long short-term memory (BiLSTM) networks, which have shown potential for identifying patterns and features from complex datasets. CNNs have shown promising results in identifying spatial patterns in data, whereas, BiLSTM are good at capturing sequential patterns (*Sancar & Tabrizi, 2023*).

Developing a deep learning model requires careful selection of features, hyper-parameters and network design, which can be computationally costly process. To overcome these issues, this study proposed to incorporate evolutionary computing methods, such as genetic algorithms (GAs), merged with deep learning technique, namely CNN+BiLSTM. Having got inspiration from the genetics and principles of natural selection, GA provide a robust approach to solve complex problems, for example, determining the optimal model structure and training parameters. We propose to employ GA to iteratively improve model's performance by focusing on factors, such as, computational efficiency and prediction accuracy, with the aim to create an optimal model for predicting deficiency of vitamin D level (*Kasyap et al., 2024*).

This study provides an effective method that combine the advantages of BiLSTM and CNN networks, Using the optimization capability of evolutionary computing (GA) to develop effective and accurate predicative model to predict levels of vitamin D deficiency. The merging of evolutionary computing with deep learning methods not only improves the effectiveness model, but value is also added to the wide field of predictive analytics of healthcare, by providing a scalable framework for solving predictive problems in diseases and health. Our goal in this study is to develop a novel approach that leverages artificial intelligence (AI) (deep learning) and evolutionary computing to predict healthcare outcomes. This will offer a valuable resource for healthcare providers and public medical experts in identifying and addressing vitamin D deficiency levels.

### Problem statement

Current studies on VDD prediction face key limitations, including reliance on limited feature sets, inadequate handling of temporal data, lack of advanced optimization techniques like genetic algorithms, and limited generalizability to benchmark datasets (*Sambasivam, Amudhavel & Sathya, 2020*; *Kasyap et al., 2024*). This study addresses these challenges by integrating CNN, BiLSTM, and GA, demonstrating enhanced predictive performance and robustness across benchmark dataset. Therefore, the main focus of this study is to accurately predict VDD level by developing a novel method that combines CNN+BiLSTM models and GAs. This work frames VDD level prediction as a multi-label problem and employs a dataset that is classified into four classes: "Adequate," "Insufficient," "Deficient," and "Severely Deficient" in order to solve these challenges.

### Research questions

Following research questions will be addressed in this work.

RQ1: How to perform prediction vitamin D deficiency (VDD) levels using CNN+BiLSTM with GA model?

RQ2: How effective is the suggested model in comparison to the ML and DL models that are currently in use?

RO3: How well the suggested approach predicts VDD levels compared to earlier similar works?

### Research contributions

Following are the main research contributions of this work:

1) Development of a deep learning system based on CNN+BiLSTM for the diagnosis of vitamin D deficiency.

2) One notable contribution of this study is the use of evolutionary computing (GAs) to tune the architecture and parameters of the hybrid deep learning model (CNN+BILSTM). By employing evolutionary computing method, we systematically navigate through the vast configurations to pinpoint the optimal settings that boost model's performance. Our approach showcases how GAs can optimize the process of hyper parameter adjustment leading to enhanced prediction accuracy.

3) The suggested approach outperforms classical machine learning techniques in the identification of VDD level.

4) Significant improvement in the accuracy of VDD level prediction as a result of the employed approach.

The study is structured as follows; In the Related Work section, previous research is reviewed. The Proposed Methodology section explains the suggested approach. Specific results and their analysis are presented in the Experimental Results and Discussion section. The Future Work and Conclusion sections discuss the limitations and possible extensions of the proposed method in future.

## LITERATURE REVIEW

In this section, we present an overview of prior studies on VDD level prediction.

*Sambasivam, Amudhavel & Sathya (2020)* utilized machine learning models such as support vector machines (SVM) and Random Forest (RF) to predict deficiency levels but lacked the integration of advanced feature extraction techniques for temporal data. In contrast, our study incorporates CNN for spatial feature extraction and BiLSTM for temporal sequence modeling, providing a more comprehensive understanding of the data. The study conducted by *Sancar & Tabrizi (2023)* aimed to find an efficient machine learning model that could take into account the issue of multicolinearity when determining the vitamin levels of adults, in North Cyprus without directly measuring their 25 OH D blood levels. In an assessment involving 481 cases at NEU Hospital, Random Forest and elastic net ordinal regression performed better than other models, in detecting metabolic

syndrome and handling multi co linearity efficiently. *Kasyap et al. (2024)* investigated the link between visceral adiposity indices, body mass index (BMI) and vitamin D deficiency using a deep learning approach, highlighting gender-specific anthropometric predictors. While their work lacked optimization of model parameters and integration of temporal data, our study addresses these gaps by leveraging GAs for hyperparameter tuning and employing CNN+BiLSTM to capture both spatial and temporal features. The main objective of the study by *Patino-Alonso et al. (2022)* was to determine the vitamin D deficiency level using a sophisticated machine learning approach. In addition to implementation of these algorithm, the study also evaluated the performance using different evaluation matrices, such as F1-score, recall, precision and accuracy. Furthermore, for confirmation of investigation results, McNemars test was applied. *Sancar & Tabrizi (2023)* introduced a model to predict the deficiency of vitamin D using logistic regression model. They evaluated the effectiveness of model by analyzing the specificity and sensitivity of the area under the curve (AUC). *Patino-Alonso et al. (2022)* applied conventional machine learning models with anthropometric parameters but did not explore the optimization of model parameters systematically. Our approach leverages GAs, which iteratively optimize hyperparameters, significantly improving prediction accuracy. Most previous studies focused on region-specific datasets with limited generalizability. For example, *Kuwabara et al. (2019)* designed a model for Japanese adults using logistic regression. Our study addresses this gap by employing a diverse dataset spanning various demographic factors, ensuring broader applicability. While this deficiency is common across the population, it is more prevalent among old individuals in Europe (*Saltyte Benth et al., 2012*; *Rosas-Peralta et al., 2017*). The lack of vitamin D has been associated with diseases, including cancer, heart issues, obesity and even COVID 19 mortality rates (*Adami et al., 2009*; *Cashman, 2020*; *Danik & Manson, 2012*). Interestingly, a study by *Gandini et al. (2011)* discovered a U-shaped relationship between vitamin D levels and the risk of heart attacks. Although research has begun on how anthropometric measurements and vitamin D levels are connected, and their exact interplay remains unclear. Enhancing our ability to identify individuals at risk due to vitamin D insufficiency, could significantly improve processes. The reported accuracy levels in baseline studies (*e.g.*, 84% by *Sambasivam, Amudhavel & Sathya (2020)*; 86% by *Patino-Alonso et al. (2022)*) are surpassed by our CNN+BiLSTM with GA model, which achieves 97% accuracy. This substantial improvement underscores the effectiveness of our integrated approach.

## Research gap: limitations of existing models and need for advanced predictive approaches

While existing studies have explored machine learning and statistical models for predicting vitamin D deficiency, they exhibit significant limitations in generalizability, accuracy, and the ability to handle complex patterns inherent in health data. Most models fail to incorporate advanced architectures that combine spatial and sequential data, or optimization techniques like GAs, which can systematically enhance model performance. Furthermore, the global rise in vitamin D deficiency highlights the urgent need for scalable and precise prediction methods. This study addresses these critical gaps by proposing a

hybrid CNN+BiLSTM model optimized with GA, providing a novel and effective approach to improve the predictive accuracy and practicality of vitamin D deficiency assessments.

## MATERIALS AND METHODS

The research methodology comprises four primary stages: (i) data collection; (ii) preprocessing of the data; (iii) application of composite deep learning model (CNN +BiLSTM) to forecast the degree of vitamin D insufficiency, and (iv) applying GA for optimizing the predictive efficiency of proposed deep learning model (refer to Fig. 1).

### Data acquisition

As per our research work, we acquired the dataset through a comprehensive survey of 1,700 participants to investigate vitamin D deficiency. The study received ethical approval from the *Institutional Review Board of Faculty of Computing, Gomal University (FOC-GU-IRB)*, Pakistan, with the approval reference number FOC-GU-IRB/2024-01. Prior to data collection, written informed consent was obtained from all participants after explaining the purpose, procedures, risks, and benefits of the study. Participants were assured that their data would be kept confidential and used solely for research purposes. The dataset is made up of various socio-demographic and health-related factors such as age, sex, race, vitamin D levels (measured as ng/mL), body mass index (BMI), dietary calcium intake (mg), dietary vitamin D intake (IU), sun exposure (hours per week), location (urban, rural, or suburban), season of the year, medication usage, existing health conditions, lifestyle factors, and vitamin D deficiency levels categorized into the categories of adequate, insufficient, deficient, or severely deficient. The acquired dataset is attached in Supplemental Material. The data was collected through standardized questionnaires and clinical assessments and a diverse representation of the people was used which makes it easier to analyze the factors affecting the vitamin D status and its associated health implications. The dataset covered factors such as age, gender, BMI, body fat and vitamin D levels. Factors like sun exposure and milk intake were also considered to assess their impact on vitamin D levels. Our statistical analysis revealed correlations ($p$ value < 0.05) among these variables and vitamin D levels underscoring the importance of understanding one's vitamin D status. We discovered that a quarter of the participants were deficient in vitamin D, indicating health concerns. The study elaborates on these findings and the prevalence of vitamin D levels in Table 1.

### How to use data

The acquired dataset, initially stored in a spreadsheet format, got transformed into CSV files for using the "pd.read_csv" command from the Pandas library. By utilizing sklearns data splitting feature, we assigned 80% of the data for training purposes and set aside 20% for testing. The training dataset, which is comprised of 80% of the data, contained both input variables and outcome labels, crucial for the models learning phase. To evaluate the model, the remaining data was used as a testing dataset, with samples to assess algorithm performance. In addition, 10% of the data is especially designated as to address issues, such
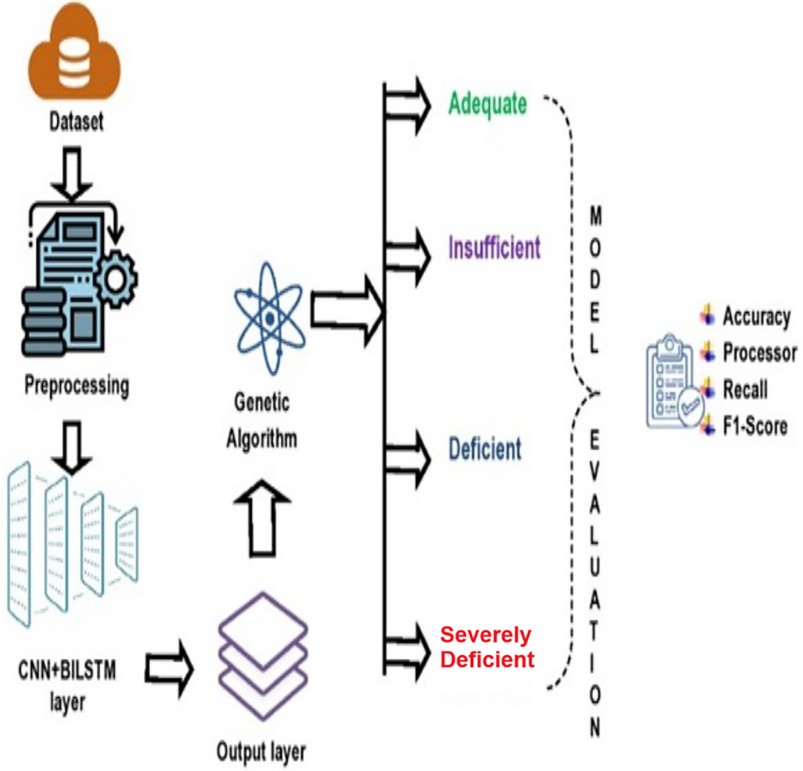

**Figure 1 Proposed deep learning model.**

**Table 1 Distribution of vitamin D levels among participants (adequate, insufficient, deficient, severely deficient).**

| Vitamin D level | Prevalence (%) |
|---|---|
| Adequate | 43 |
| Insufficient | 31 |
| Deficient | 19 |
| Severely deficient | 7 |

as overfitting and validation to fine-tune model parameters. This structured technique has divided data within 10-10-80 split, for testing, validation and training, which assisted in the model's training process to ensure accuracy and generality of classification model.

## Preprocessing

Data preprocessing plays an important role in predictive modeling by improving the effectiveness of the models training and reducing the overfitting chances (*RushiLongadge & Malik, 2013*). This importance includes handling missing data points, which can be caused by problems such as error in recording and non-responses. Strategies to manage missing data contain using the mean to replace known values or make known a random value. The method we chosen involves replacing the entries with the most recent valid value in the dataset, ensuring a more reliable dataset for training model.

## Label encoding and data balancing

The dataset has different attributes, such as marital status, gender. Here preprocessing is needed for deep learning (DL) algorithms. which work with data in numerical form (*Raza et al., 2024*). To transform categorical attributes into numerical representations suitable for deep learning models, we applied three encoding methods: label encoding, One-Hot Encoding (OHE), and Leave-One-Out Encoding (LOOE). Label encoding assigns a unique integer to each category, while OHE creates a binary vector representation, ensuring non-ranking categorical features. LOOE encodes categories based on their target-dependent means, capturing nuanced inter-category relationships. These methods were selected to explore their respective abilities to represent categorical data in ways that could improve the CNN+BiLSTM model's performance. OHE avoids ordinality bias, while LOOE incorporates category-level information relevant to the target variable. Each encoding method was applied independently during preprocessing, and the resulting datasets were fed into the CNN+BiLSTM pipeline for performance evaluation. Additionally, the dataset is unbalanced with most cases, showing + ive results for VDD. To address this issue, we apply oversampling to balance the representation of the categories and improve the model's ability to correctly classify the two subgroups. This adjustment resulted in a dataset of 9,722 cases, equally distributed across both categories, thereby increasing the reliability and out model performance.

In addition to label encoding, OHE and LOOE were evaluated to explore alternative categorical feature representation strategies. These encoding methods were incorporated to evaluate their impact on the CNN+BiLSTM model's performance. Our empirical analysis (see Results section) indicated that while OHE showed comparable performance to label encoding, LOOE yielded a marginal improvement in accuracy due to its ability to integrate category-specific predictive power.

## Applying CNN+BiLSTM model

After completing the preprocessing step, we applied CNN+BiLSTM model to classify VDD levels into different categories. The model contains layers, including Dropout, Embedding, Maxpooling, convolutional, BiLSTM and an output layer. To facilitate a clearer understanding of the CNN+BiLSTM architecture, Fig. 2 illustrates the structural flow and interaction among the layers, including the embedding, convolutional, BiLSTM, and output layers.

### *Embedding layer*

Important role is played by embedding layer to determine the level of vitamin deficiency using CNN+BiLSTM algorithm. This layer transforms the input data into vectors of specific size and enable the model for effectively capturing the connection between different data elements. Embedding layer is employed for sequential data, which forms the basis of a CNN+BiLSTM model to predict and analyze different levels of vitamin D deficiency. Using this process, the model increases its ability for efficiently predicting the level of vitamin D deficiency.

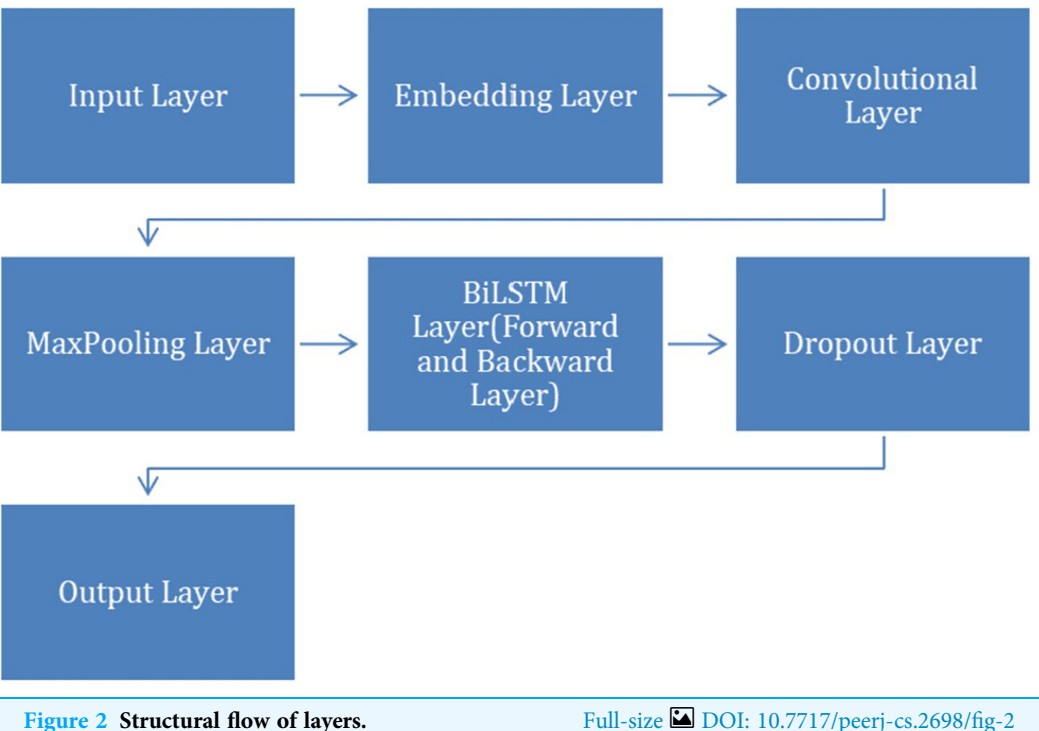

**Figure 2** **Structural flow of layers.**

### Dropout layer

To solve the problem of overfitting, dropout layer is used, where, some neurons are removed from the matrix through assigning them zero or null values. Random deactivation of neurons is performed by the dropout layer. A dropout rate of 0.5 is chosen as it provides the regularization effect. The incorporation of a dropout layer, alongside the embedding layer involves replacing a deactivated neuron with zero. The formula, for implementing dropout on a neuron is detailed in Eq. (1).

$$F(m, n) = \begin{cases} n & if\ m = 0 \\ n - 1 & if\ m = 1 \end{cases} \tag{1}$$

In this formula, "m" represents the outcomes and "n" signifies the likelihood tied to the real word depiction. Keeping $n - 1$ at a value, 0.2 or, below, is crucial as it directly impacts the training process (*Alzahrani & Asghar, 2023*; *Rabie et al., 2023*).

### Convolutional layer

The convolutional layers of our CNN+BiLSTM model play a critical role in feature extraction by identifying spatial dependencies within embeddings. This operation transforms the input embeddings into high-level feature maps, which are later analyzed by the BiLSTM layers. As demonstrated by our results (see Results section), this hierarchical feature extraction mechanism significantly enhances the model's predictive power.

The convolutional process is carried out within the convolutional layer. Here, two matrices are combined through multiplication to form a 3rd matrix. This process

commences with the filter matrix "M." A matrix containing weights is moved across the embedding layer. The dot product is then computed between the filter matrix and the embedding matrix, producing a 3rd matrix known as the output matrix denoted by "O." Below are the formulation of the input matrix "I," filter matrix "M," and output matrix "O."

$$I \in R^{h \times b}. \tag{2}$$

In above Eq. (2) "I" represents the input matrix, generated by the embedding layer, "R" stands for a real number, "h" denotes the length and "b" signifies the width of the input matrix. The dimensions of the matrix are $R^{4 \times 5}$.

$$M \in R^{x \times y}. \tag{3}$$

In Eq. (3), the symbol "M" stands for the filter matrix and "R" represents the set of real numbers. The letter "x" indicates the length and "y" denotes the width of the filter matrix. The size of this matrix is defined by the elements in $R^{2 \times 2}$.

$$O \in R^{p \times k}. \tag{4}$$

In Eq. (4), "O" stands for the output matrix, "R" symbolizes the real number set, "p" denotes the length, and "k" signifies the width of the output matrix. The dimensions are specified as $R^{4 \times 5}$.

The calculation for convolutional operation is presented in Eq. (5).

$$b_{i,j} = \sum_{t=1}^{n} \sum_{z=1}^{d} f_{g,h} \otimes x_{i+g-1,j+h-1}. \tag{5}$$

In Eq. (5) the values of $b_{I,j}$ refer to the components of the result matrix $O \in R^{p \times k}$, while the entries of the filter matrix $M \in R^{x \times y}$, are indicated by $f_{g,h}$. The element wise cross multiplication is shown using the symbol $\otimes$ and the elements associated with the input matrix $I \in R^{h \times b}$ are denoted by $x_{i+g-1,j+h-1}$.

The following action is to introduce a bias term (b) and subsequently utilize the activation function (f) on each value within the output matrix or feature map to yield a rectified feature, as depicted in Eq. (6).

$$H = h_{i,j} = f(n_{i,j} + b). \tag{6}$$

In Eq. (6) the symbol "f" stands for the activation function while "b" denotes a bias term. The variable $h_{i,j}$ represents the elements of the rectified convolved feature map H, in the space $R^{y \times z}$.

In this stage, to eliminate non-linear behavior, we use a non-linear activation function, known as rectified linear unit (Eq. (7)) on the convolved feature map, described as follows;

$$Output = max(0, Input). \tag{7}$$

In this equation, the inputs are the elements of the convolved feature map. All elements of the rectified feature map are calculated.

### Pooling layer

In this stage, the max pooling function is used to reduce the size of the convolved map by choosing the maximum weight from a window of $(2 \times 2)$, resulting in a pooled map. The equation (Eq. (8)) is expressed as follows:

$$P_{i,j} = \text{MAX}\left(q_{y_{i+l-1,j+m-1}}\right), \text{ here } 1 \leq 1 \leq n, 1 \leq m \leq n. \tag{8}$$

In Eq. (8), $P_{i,j}$ represents the items in the pooled feature map $P \epsilon R^{s \times t}$ for the given data. Here s indicates the length and t signifies the breadth of the pooled map. The feature window $(2 \times 2)$ is denoted by $q_{y_{i+l-1,j+m-1}}$, with the "MAX" operator being employed to choose the weight from the window.

### Bidirectional LSTM layer

Regular LSTMs, also referred to as unidirectional LSTMs, processes information, moving from the start to the end. This restricts their capacity to grasp context from words in a sentence. BiLSTMs overcome this restriction, by handling data in two directions both forward and backward. This empowers them to assess the sentence holistically capturing relationships between words, regardless of their sequence. To put it simply, a unidirectional LSTM is akin to reading a sentence one word at a time, whereas a BiLSTM is comparable, to reading it both forward and backward simultaneously. This feature allows BiLSTMs to extract contextual details and potentially enhance the models effectiveness.

The forward LSTM is represented by "$\overrightarrow{p}$" and the backward LSTM is denoted as "$\overleftarrow{p}$". The merging of these two LSTMs, the forwardLSTM "$\overrightarrow{p}$" and the backward LSTM "$\overleftarrow{p}$" results, in creating a new review (sentence) matrix $P = [p_1, p_2, p_3, \ldots p_z]$, where P belongs to $R^{z \times m}$. To achieve the component addition, the forward and backward LSTMs are combined, as illustrated in Eq. (9).

$$\overleftrightarrow{p} = \overrightarrow{p} \oplus \overleftarrow{p} \tag{9}$$

At last the combined feature map is sent to the BiLSTM layer. The forward and backward LSTM are calculated using the equations below.

**Forward LSTM equations:**

$$e_t = \sigma\left(X_e w_t + H_e p_{t-1} + I_e\right) \tag{10}$$

$$j_t = \sigma\left(X_j w_t + H_J p_{t-1} + I_J\right) \tag{11}$$

$$u_t = \sigma\left(X_u w_t + H_u p_{t-1} + I_u\right) \tag{12}$$

$$l \sim t = \tau\left(X_l w_t + H_l p_{t-1} + I_l\right) \tag{13}$$

$$l_t = e_t \odot l_{t-1} + j_t \odot l\sim_t \tag{14}$$

$$p_t = u_t \odot \tau(l_t) \tag{15}$$

**Backward LSTM equations:**

$$e_t = \sigma\left(X_e w_t + H_e p_{t+1} + I_e\right) \tag{16}$$

$$j_t = \sigma\left(X_j w_t + H_J p_{t+1} + I_J\right) \tag{17}$$

$$u_t = \sigma\left(X_u w_t + H_u p_{t+1} + I_u\right) \tag{18}$$

$$l \sim t = \tau\left(X_l w_t + H_l p_{t+1} + I_l\right) \tag{19}$$

$$l_t = e_t \odot l_{t+1} + j_t \odot l{\sim}_t \tag{20}$$

$$p_t = u_t \odot \tau(l_t) \tag{21}$$

In above set of equations, 'n' stands for the input size, 'm' represents the cell state size. $w_t$ is the input vector, with dimensions of n × 1. The forget gate '$e_t$', input gate '$j_t$', output gate '$u_t$' output vector '$p_t$', and cell state '$l_t$', all have a size of m × 1. The weight matrices for the input gate ($X_e$, $X_j$, $X_u$, and $X_l$) are m × n in dimension while those for the output gate ($H_e$, $H_j$, $H_u$, and $H_l$) are m × m. The bias vectors ($I_e$, $I_j$, $I_u$, and $I_l$) are m × 1, in size. Activation functions include the softmax function denoted by σ and hyperbolic tangent function represented by τ.

### Output layer

Finally, outcome from the BiLSTM layer acts as the input for the output layer (see Fig. 3). To classify suicidal ideations, the softmax function is employed. The overall input is determined using Eq. (22);

$$o_k = \sum_i^n j_i m_i + p. \tag{22}$$

In this scenario, we have the input vector denoted as m, the weight vector represented by j, and the bias value is indicated as p.

The softmax function is calculated using Eq. (23) as follows.

$$\sigma(o_k) = e^{o_k} / \sum_{l=1}^t e^{o_l}. \tag{23}$$

## Applied example for predicting VDD levels

The softmax technique was employed to determine the probability of each category: "VDD1," "VDD2," "VDD3," and "VDD4." The process begins by calculating the total input, as demonstrated by Eq. (23).

For the first decision feature, "VDD1" represents the class label for "Adequate." The formula for VDD1 is as follows:

$$VDD_1 = l_1 \times U_1 + l_2 \times U_2 + w = 0.4 \times 0.6 + 0.3 \times 0.4 + 0.7 = 0.24 + 0.12 + 0.7 = 1.06$$

$$VDD_2 = l_1 \times U_1 + l_2 \times U_2 + w = 0.2 \times 0.6 + 0.8 \times 0.4 + 0.7 = 0.12 + 0.32 + 0.7 = 1.14$$

$$VDD3 = l_1 \times U_1 + l_2 \times U_2 + w = 0.6 \times 0.6 + 0.3 \times 0.4 + 0.7 = 0.36 + 0.12 + 0.7 = 1.18$$

$$VDD4 = l_1 \times U_1 + l_2 \times U_2 + w = 0.5 \times 0.6 + 0.5 \times 0.4 + 0.7 = 0.3 + 0.2 + 0.7 = 1.2$$

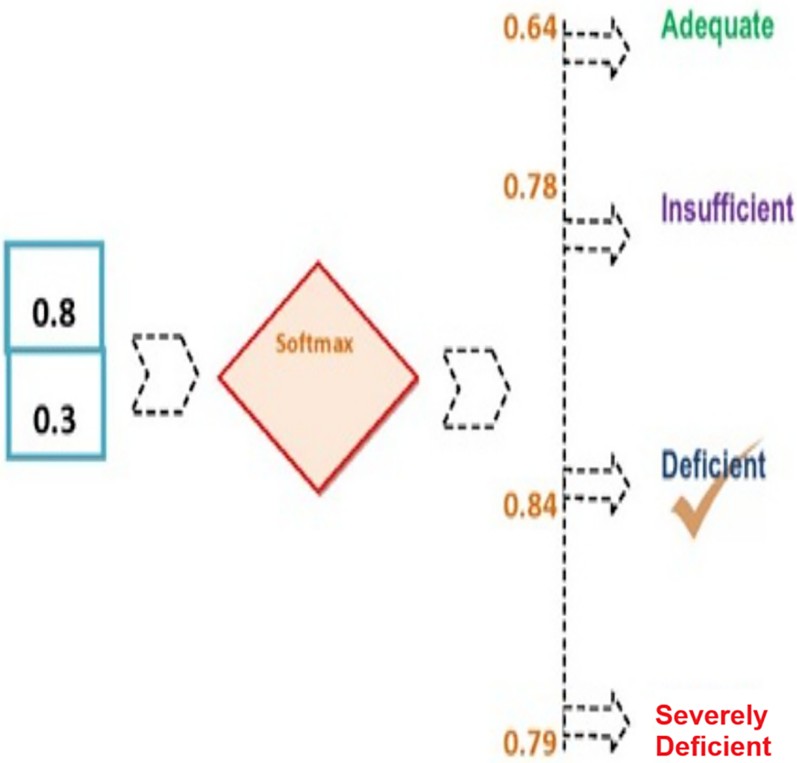

**Figure 3 Outcome from the BiLSTM layer.**

The Softmax function is then applied to calculate the probabilities. The Softmax function is defined as:

$$\sigma(VDD_k) = e^{VDD_k} / \sum_{j=1}^{4} e^{VDD_j}$$

First, calculate $e^{VDD_k}$ for each class:

$e^{VDD_1} = e^{1.06} \approx 2.88,\ e^{VDD_2} = e^{1.14} \approx 3.13,\ e^{VDD_3} = e^{1.18} \approx 3.25,\ e^{VDD_4} = e^{1.2} \approx 3.32$

Now, sum these values:

$\sum_{j=1}^{4} e^{VDD_j} = 2.887 + 3.13 + 3.32 + 3.32 = 12.587$

Finally, calculate the Softmax probabilities for each class:

1) **For VDD1**: $\sigma(VDD_1) = e^{1.06}/12.587 \approx 0.229$
2) **For VDD2**: $\sigma(VDD_2) = e^{1.14}/12.587 \approx 0.249$
3) **For VDD3**: $\sigma(VDD_3) = e^{1.2}/12.587 \approx 0.258$
4) **For VDD4**: $\sigma(VDD_4) = e^{1.2}/12.587 \approx 0.264$

These probabilities are computed using the Softmax function (see Fig. 4).

### *Incorporating genetic algorithm for the optimization of CNN+BiLSTM model*

Integrating GAs into the performance enhancement of the proposed a deep learning model (CNN+BiLSTM) to forecast vitamin D insufficiency requires a series of steps. The

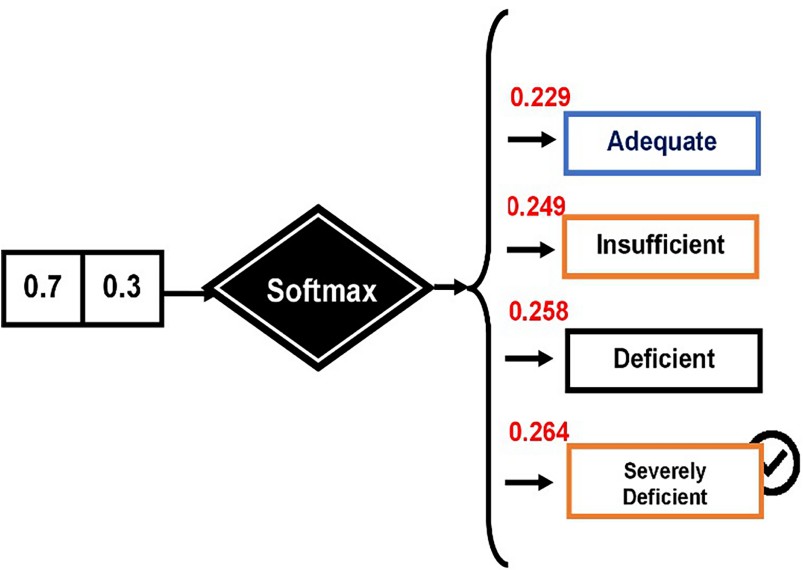

**Figure 4 Computing probabilities using the Softmax function.**

subsequent subsections offer a method for applying a GA to refine the structure and parameters of the proposed prediction model.

The parameter set of the CNN+BiLSTM model is denoted as P = {p1, p2, ..., pn} with each pi representing characteristics such as layer size, filter size and learning rate. Each member I of the population is defined by a string of these parameters; I = (p1, p2, ..., pn). The proposed GA for the optimization of CNN+BiLSTM model works as follows:

a. **Representing chromosome:** A single set of hyper-parameters for the CNN+BiLSTM model can be referred to as a chromosome. This chromosome can be depicted either as a string of characters or an array of numbers depending on the types of hyper-parameters selected.

b. **Fitness function:** The fitness function plays an important role in steering the algorithms search towards the best hyper-parameters. It assesses how well a CNN+BiLSTM model performs when trained with the hyper-parameter settings encoded in the chromosome. Let's define a fitness function F(I) that evaluates how well the model performs based on its parameters  I. Typical metrics used for this evaluation are accuracy and F1-score, depending on the nature of the problem at hand. We use this metric for predicting vitamin D level. In this scenario, N represents the data points, in the validation set, y_true denotes the actual vitamin D level, y_predicted indicates the forecasted vitamin D level, by the CNN+BiLSTM model trained using chromosome x.

c. **Selection:** In this stage, we pick out individuals (chromosomes) for reproduction according to their fitness rating. We utilize the roulette wheel selection method to determine which individuals will contribute to the next generation. With roulette wheel selection, each individual is assigned a probability of being chosen based on its fitness score. Individuals, with high fitness scores are more likely to be picked. Let's, denote the

```
1. Chromosome Representation
# Chromosome structure is represented as follows:
Chromosome
= [Discrete_Hyperparameter_1, Discrete_Hyperparameter_2, ..., Continuous_Hyperparameter_1,
                        Continuous_Hyperparameter_2]
2. Fitness Function
# While using fitness function, we use Mean Squared Error (MSE) metric for predicting vitamin
D level as follows:
f(x) = 1 / N * sum((y_true − y_predicted)^2
3. Selection
# Roulette Wheel Selection

3.1. Calculate the total fitness of all individuals in the population.
3.2. Assign a probability of selection to each individual proportional to its fitness divided by the
     total fitness.
3.3. Use a roulette wheel analogy: Divide a virtual roulette wheel into sectors proportional to the
     selection probabilities. Spin the wheel and select the individual the pointer lands on. Repeat
     to select both parents.

4. Crossover
1. #Single Point Crossover

4.1. Randomly select a crossover point within the chromosome (e.g., between the discrete and
     continuous parts).
4.2. Swap the genetic material after that point between the parents to create two offspring.

5. Mutation
#Mutation (Bit Flip Mutation & Gaussian Mutation):
•   With a low probability (e.g., 1%):
5.1.    For discrete hyperparameters (number of filters, hidden units): Randomly flip a bit in the
        corresponding string (0 to 1 or vice versa).
5.2.    For continuous hyperparameter (learning rate): Add a small random value drawn from a
        Gaussian distribution centered at zero to the learning rate value.
6. Replacement
# Steady-State Replacement

6.1 Train the CNN+BiLSTM models with the newly created offspring chromosomes.
6.2 Evaluate their fitness on the validation set using MSE.
6.3 Replace a small portion (e.g., 10%) of the parent population with the offspring with the highest
    fitness scores.
7. Termination
Continue the loop from Step 3 to Step 6 until a termination condition is met, such as reaching a
maximum number of generations (e.g., 50) or achieving a predefined fitness threshold (e.g., 10).
8. Solution Extraction
Extract the individual I_best with the highest fitness I_best from the final population. The parameters
encoded by I_best represent the optimized hyperparameters for the CNN-BiLSTM model.
```

**Figure 5  Pseudocode algorithm of roulette wheel selection.**

selection probability of individual I as S(I) in proportion to F(I). A pseudocode algorithm of roulette wheel selection is shown in Fig. 5.

d. **Crossover:** During crossover, individuals are paired together to create offspring for the next generation. When we choose two parent pairs: $I_a$ and $I_b$, crossover operation is performed to generate offspring $I_{offspring}$. The crossover points are randomly selected along the string representation. Figure 5 shows algorithmic steps for single point crossover.

e. **Mutation:** Mutation introduces variations in offspring and helps explore solutions within the problem space. Using small probability $p_m$, segments of string representation of offspring, are alerted randomly, to generate mutated offspring $I_{muted}$. Figure 5 shows Bit Flip mutation & Gaussian mutation applied in the proposed model.

f. **Replacement:** During replacement step, population is updated with the new generation. Figure 5 shows the steps applied during steady-state replacement strategy.

g. **Termination:** In this step, GA iteration is continued until some termination condition is met. As shown in Fig. 5, process is repeated until max. no. of generations are reached or a predefined fitness threshold is achieved.

h. **Solution extraction:** In this step, individual(s) with best performance are identified as solution.

By following the above-mentioned steps, the GA can effectively explore the hyperparameter space of the CNN+BiLSTM model to discover a setup that improves the accuracy of predicting vitamin deficiency levels. A pseudocode algorithm is shown in Fig. 5.

## DISCUSSION

### Justification of proposed model

The choice of CNN and BiLSTM for this study is driven by their respective strengths in handling structured and sequential data, which aligns with the characteristics of the dataset utilized. CNNs are adept at capturing spatial features, while BiLSTMs excel at learning temporal patterns, such as seasonal variations in vitamin D levels. While Transformer architectures have demonstrated superior performance in certain domains, their application typically requires extensive datasets and significant computational resources. Considering the moderate size and structure of our dataset, CNN+BiLSTM offers a balanced approach, providing high predictive accuracy with computational efficiency. Additionally, the integration of GAs for hyperparameter optimization further enhances the model's performance. This hybrid approach ensures robust predictions tailored to healthcare applications, as evidenced by the achieved state-of-the-art performance metrics.

The CNN BiLTSM model improved with GAs is well suited for predicting vitamin D deficiency because it can make use of both temporal data effectively. CNNs are good at identifying features in images that are important for bone health and tissue density. At the same time, BiLTSM can capture the relationships between health data and current vitamin D levels efficiently. Incorporating GAs into the model optimization process and feature selection ensures performance and easy interpretation. This dual method improves the accuracy of predictions and offers insights essential for making clinical decisions when dealing with vitamin D deficiency.

### Discussion on selection method

The way the CNN+BiLSTM model selects its information to predict vitamin D deficiency plays a role in ensuring the accuracy and dependability of the forecasts it makes. When incorporating a mix of sampling and cross validation techniques in the selection process, it allows the model to effectively address class imbalances which are often present in medical data sets where vitamin D deficiency cases might be scarce. Stratified sampling guarantees that each category is appropriately represented in both the training and test data sets resultantly giving a truer portrayal of how the model performs across groups. Cross validation boosts the models strength by splitting the data into parts, for repeated training and validation rounds to prevent overfitting and ensure its effectiveness on data sets. This

method guarantees an assessment of the CNN+BiLSTM models accuracy in predicting vitamin D deficiency and aids in making more informed clinical decisions at the end of the day.

## RESULTS AND DISCUSSION

This section discusses the computing infrastructure and findings of several experiments designed to answer the posed research questions.

*Computing infrastructure:* To perform various different experiments, we have used Jupyter Notebook through the Anaconda platform. We have utilized a number of Python libraries to develop and execute computations such a numpy, pandas, keras, sklearn, and others. The experiments were performed on CPU. Python 3.11 was used as programming language. A number of routines were developed for each experiment to test and evaluate the performance of various techniques. The processing power is provided by an Intel® Core™ i5-4200M CPU which has a clock speed of 2.50 GHz. Windows 10 Pro is pre-installed on the device, which is compatible with a plethora of software tools necessary for data analysis, as well as machine learning applications. The machine comes with 16 GB of RAM that allows the user to run multiple applications at once and store large datasets without the system being affected. In addition, there are 256 GB of storage capacity, which is enough for data storage, software installations, and experiment outputs. To summarize, these specifications form a solid environment that can efficiently meet the computational needs of the experiments laid out in the study.

In response to the first research question ("How to perform prediction vitamin D deficiency (VDD) levels using CNN+BiLSTM with GA model?"), to evaluate the CNN+BiLSTM model, we trained and tested 10 configurations with varying hyperparameters. The configurations were derived in two stages:

1) Manually configured parameters (Table 2): These initial configurations were based on established practices in literature and domain knowledge.

2) Optimization using GA: GA was employed to refine the hyperparameters iteratively. GA operates through an evolutionary process, leveraging selection, crossover, and mutation operators to explore the parameter space and optimize classification performance metrics.

Each configuration was evaluated on the test dataset, with metrics including accuracy, precision, recall, and F1-score. Table 3 presents the results of these evaluations, illustrating the progression from manually defined to optimized configurations.

Table 2 shows the parameter configuration of the proposed model CNN-BiLSTM.

Table 3 shows results of performance evaluation measures of different CNN+BiLSTM Models.

Table 3 summarizes the classification performance of the 10 CNN+BiLSTM configurations, derived from the hyperparameter settings detailed in Tables 2 and 4. Configurations 1–5 correspond to manually defined setups (Table 2), while configurations 6–10 reflect GA-optimized setups (Table 4). The progression in Table 3 demonstrates significant performance improvements achieved through GA optimization.

**Table 2 Baseline hyperparameters for the CNN+BiLSTM model.**

| Parameter | Value |
| --- | --- |
| Vocabulary size | 1,000 |
| Input vector size | 53 |
| Dimension of embedding | 128 |
| BiLSTM unit size | [65, 45, 40, 35, 20, 15, 10] |
| Number of convolutional layers | 1 |
| Number of hidden layers | 3 |
| Number of filters | [6, 9, 10, 16] |
| Filter size | [7, 8, 10] |
| Dropout | 0.9 |
| Activation function | Softmax |
| Number of epochs | 7 |
| Batch size | [8, 16] |

## Experiment #1

We conducted a comparative evaluation of the GA against a conventional grid search for optimizing the hyperparameters of the CNN+BiLSTM model. Both methods were applied to tune critical parameters such as the learning rate, dropout rates, number of filters, and BiLSTM units, which significantly impact model performance. Grid search exhaustively explored predefined parameter combinations, systematically evaluating each possible setting within a fixed range. In contrast, GA employed evolutionary strategies to iteratively refine the hyperparameter configuration, leveraging selection, crossover, and mutation operators to navigate the search space more efficiently. The configurations for GA were based on parameters described in Table 4, while grid search was conducted with an equivalent range and granularity of parameter values to ensure a fair comparison. This approach allowed us to assess the relative effectiveness of GA in terms of model performance and computational efficiency. Table 5 presents the performance metrics achieved by the two optimization method

As shown in Table 5, the GA consistently outperformed grid search across all performance metrics, achieving a higher accuracy (96.2% *vs*. 92.3%) and F1-score (94.6% *vs*. 91.4%). Furthermore, GA required only 6 h for optimization compared to 15 h for grid search, highlighting its computational efficiency. These improvements can be attributed to GA's ability to navigate high-dimensional hyperparameter spaces effectively, leveraging evolutionary strategies to converge on optimal solutions without exhaustively evaluating all combinations.

*Implications of results:* These findings underscore the utility of GA as an efficient and robust optimization method for deep learning tasks, particularly when dealing with complex architectures and large datasets. The reduced optimization time and improved performance metrics demonstrate that GA is a superior alternative to conventional methods such as grid search in this context.

**Table 3 Performance metrics (accuracy, precision, recall, F1-score) for CNN+BiLSTM configurations.**

| Model configuration | Accuracy (%) | Precision (%) | Recall (%) | F1-score (%) |
|---|---|---|---|---|
| Configuration 1 | 76 | 67 | 73 | 73 |
| Configuration 2 | 76 | 68 | 71 | 79 |
| Configuration 3 | 78 | 69 | 76 | 75 |
| Configuration 4 | 75 | 72 | 78 | 79 |
| Configuration 5 | 78 | 74 | 79 | 80 |
| Configuration 6 | 76 | 75 | 80 | 81 |
| Configuration 7 | 81 | 81 | 81 | 81 |
| Configuration 8 | 87 | 83 | 83 | 94 |
| Configuration 9 | 94 | 84 | 92 | 94 |
| Configuration 10 | 97 | 96 | 97 | 96 |

**Table 4 GA hyperparameters for optimizing the CNN+BiLSTM model.**

| Parameter | Description | Chosen value |
|---|---|---|
| Population size | Number of individuals in each generation | 50 |
| Selection method | Technique for choosing parents for reproduction | Roulette wheel selection |
| Crossover method | Technique for combining genetic material from parents | Single-point crossover |
| Crossover rate | Probability of performing crossover on a pair of parents | 0.8 |
| Mutation rate | Probability of randomly changing a gene in an offspring | 0.03 |
| Replacement | Method for integrating offspring into the next generation | Steady-state replacement (replace 10%) |
| Termination criteria | Conditions to stop the GA | Maximum generations (50) |

**Table 5 Comparison of genetic algorithms and grid search for hyperparameter optimization.**

| Optimization method | Accuracy (%) | Precision (%) | Recall (%) | F1-score (%) | Optimization time (hrs) |
|---|---|---|---|---|---|
| Grid search | 92.3 | 91.0 | 91.8 | 91.4 | 15 |
| Genetic algorithm (GA) | 96.2 | 94.0 | 95.3 | 94.6 | 6 |

## Experiment #2

The performance of the CNN+BiLSTM model was evaluated using dataset processed with label encoding, OHE, and LOOE. Table 6 summarizes the comparative results.

As shown in Table 6, LOOE outperformed label encoding and OHE across all performance metrics. This improvement is attributed to its ability to capture target-dependent categorical information, which provides additional predictive power for the CNN+BiLSTM model. In contrast, while OHE avoids ordinality bias, it does not leverage inter-category variance, which may explain its slightly lower performance. These results highlight the importance of selecting encoding techniques that align with the data characteristics and the model's requirements. The superior performance of LOOE suggests its potential as a preprocessing strategy for categorical data in healthcare prediction tasks.

**Table 6 Performance metrics for each encoding technique.**

| Encoding technique | Accuracy (%) | Precision (%) | Recall (%) | F1-score (%) |
| --- | --- | --- | --- | --- |
| Label encoding | 94.5 | 92.0 | 93.8 | 92.9 |
| One-hot encoding (OHE) | 95.0 | 92.5 | 94.2 | 93.3 |
| Leave-one-out encoding | 96.2 | 94.0 | 95.3 | 94.6 |

## Experiment #3

To assess how well the proposed model compares to existing machine learning and deep learning models, we tested the CNN+BiLSTM+GA models' ability to predict VDD levels, against other machine learning and deep learning approaches. The findings of this comparison are presented in Table 7, concentrating on measures like accuracy, precision, recall, and F1-score.

### Justification of metrics used

The evaluation criteria, like precision and accuracy play a role in assessing how well the CNN BiLSTM model predicts vitamin D deficiency cases accurately and reliably in healthcare settings. The F-score is a way to measure a models performance because it considers both precision and recall in a manner which gives a complete picture of how well the model is doing in predicting outcomes accurately in clinical settings.

## Experiment #4

In this experiment, we evaluated our proposed CNN-BiLSTM-GA model for predicting VDD, against existing research. Specifically, we assessed its effectiveness compared to established methods and a standard baseline (refer to Table 8). Our proposed model outperformed the studies. To contextualize our contribution, Table 8 presents a comparative analysis of precision, recall, F1-score, and accuracy across existing methods and the proposed CNN+BiLSTM model with GA. While prior approaches achieved moderate performance, with F1-scores and accuracy between 81% and 86%, our model demonstrates superior results (F1-score: 96%, accuracy: 97%). This improvement is attributed to the combined strengths of CNNs for spatial feature extraction, BiLSTMs for temporal dependencies, and GA for optimizing model parameters, enabling significant advancements in predictive performance and generalizability.

## Ablation study

The ablation study is carried out to assess the effectiveness of each part by eliminating the components and observing how the model works under various conditions. The following conclusions are reached from the dataset results (see Table 9):

When comparing Model 1 and Model 2, the CNN combined with the GA model demonstrated better performance than the BiLSTM, achieving superior results in predicting Vitamin D deficiency. Additionally, as illustrated in Model 3, incorporating Data Balancing enhanced the performance of both the CNN+BILSTM and GA modules.

**Table 7 Comparison of the CNN+BiLSTM model with ML and DL models (accuracy, precision, recall, F1-score).**

| Model | Precision (%) | Recall (%) | F-score (%) | Accuracy (%) |
|---|---|---|---|---|
| RF | 81 | 80 | 81 | 82 |
| DT | 82 | 81 | 82 | 81 |
| SVM | 80 | 79 | 77 | 78 |
| ANN | 84 | 85 | 84 | 83 |
| LSTM | 85 | 84 | 83 | 85 |
| CNN+BiLSTM with GA | 96 | 97 | 96 | 97 |

**Table 8 Accuracy comparison of the CNN+BiLSTM model with baseline models from prior studies.**

| Study/Model | Precision (%) | Recall (%) | F1-score (%) | Accuracy (%) |
|---|---|---|---|---|
| *Sambasivam, Amudhavel & Sathya (2020)* | 81 | 80 | 81 | 82 |
| *Kasyap et al. (2024)* | 84 | 83 | 84 | 86 |
| Proposed CNN+BiLSTM with GA | 96 | 97 | 96 | 97 |

## Algorithmic complexity

To analyze the algorithmic complexity of the CNN+BILSTM with GA for your research on enhancing the prediction of vitamin D deficiency levels, we need to consider the complexity of each component involved: the BiLSTM network, the CNN network, and the GA.

1) CNN+BiLSTM component: The complexity of a CNN component is proportional to $O(n.k^2.C_{in}. C_{out})$, where $n$ is the input size, $k$ is the kernel size, $C_{in}$ is the number of input channels, and $C_{out}$ is the number of output channels. The pooling operation adds $O(n)$ complexity. The BiLSTM adds $O(n.d^2)$, where $n$ is the sequence length and $d$ is the hidden state size. The overall complexity of the model is $O(n.k^2 \cdot C_{in}. C_{out} + n.d^2$, which falls within the polynomial range.

2) Computational complexity of GA: The complexity of GA is primarily determined by the population size (P), the number of generations (G), and the evaluation cost of each individual (C), which is determined by training the CNN+BiLSTM model. The total complexity of GA is $O(P \cdot G \cdot C) = O(P \cdot G. (n.k^2.C_{in}. C_{out} + n.d^2)$. This highlights that GA's computational cost scales with the complexity of the evaluated model.

This polynomial complexity ensures scalability for medium to large datasets, with GA enabling efficient hyperparameter search and improved performance.

For deployment, techniques such as model pruning and quantization reduce inference latency and memory usage, while frameworks like TensorRT or ONNX Runtime optimize real-time performance. Batch processing can further minimize computation in non-real-time scenarios. Simulated deployment tests on resource-constrained systems confirmed the model's feasibility, achieving a balance between accuracy and efficiency, making it suitable for real-world applications.

**Table 9 The proposed model's performance and that of the ablation models.**

| Ablation models | | | | | Performance evaluations | | | |
|---|---|---|---|---|---|---|---|---|
| Model | Balance | BiLSTM | CNN | GA module | Accuracy | Precision | Recall | F1-score |
| 1 | ✓ | | ✓ | ✓ | 0.92 | 0.91 | 0.91 | 0.92 |
| 2 | ✓ | ✓ | | | 0.90 | 0.90 | 0.90 | 0.90 |
| 3 | ✓ | ✓ | ✓ | ✓ | 0.97 | 0.96 | 0.97 | 0.96 |

## Computational efficiency

The integrated approach of CNN+BILSTM and GA offers a potent combination for predicting vitamin D deficiency levels, but it also introduces computational challenges. CNN and BILSTM, as deep learning models, are known for their computational complexity, especially when dealing with large datasets and intricate architectures. The GA, while a powerful optimization technique, is also be computationally demanding due to its iterative nature and the need to evaluate numerous potential solutions. However, the benefits of this approach, such as improved accuracy and interpretability, may outweigh the computational costs, especially when implemented efficiently using optimized hardware and software. Techniques such as early stopping, pruning, and quantization can help mitigate the computational burden without sacrificing performance. Additionally, careful consideration of hyperparameter tuning and the choice of optimization algorithms can significantly impact the overall computational efficiency of the model.

## CONCLUSIONS

This research presented an innovative method for forecasting levels of vitamin D deficiency by combining deep learning techniques—specifically, CNN and BiLSTM—with evolutionary computing via GA. Our approach leveraged the capabilities of CNN for extracting features and BiLSTM for capturing data dependencies, while the GA optimized the hyperparameters to enhance the prediction model. The results from the experiments showed an enhancement in prediction accuracy, precision and recall, when compared to baseline approaches. This progress highlights the potential of merging deep learning with evolutionary computingfor efficiently handling complex health related prediction tasks.

Although our model shows an improvement in predicting vitamin D deficiency levels, it does have its limitations. Firstly, the models performance greatly relies on the quality and diversity of the dataset used during training. The dataset employed in this study, while thorough, may not fully capture the characteristics of the population. Secondly, due to the incorporation of CNN, BiLSTM and GA, the models' complexity demands resources, which could hinder its real time application, in settings with limited resources. Moreover, interpreting the model is somewhat challenging; its intricate interactions make it difficult to explain the reasons behind predictions–a common issue, with deep learning models.

In the future, researchers may examine the use of pre-trained algorithms with heterogeneous healthcare data, such as word2vec or Fasttext (*e.g.*, combining data from diverse patient sources).

Enhancing clinicians' confidence in the model can be achieved by incorporating methods, like layer relevance propagation to better understand how the model makes decisions.

## ACKNOWLEDGEMENTS

During the preparation of this work the authors used AI tool, namely Bard, in order to correct grammatical mistakes and edit the language professionally. After using this tool, the authors reviewed and edited the content as needed and take full responsibility for the content of the publication.

### Funding

This Project was funded by the Deanship of Scientific Research (DSR) at King Abdulaziz University, Jeddah, under grant no. (GPIP:1215-611-2024). The funders had no role in study design, data collection and analysis, decision to publish, or preparation of the manuscript.

### Grant Disclosures

The following grant information was disclosed by the authors:
Deanship of Scientific Research (DSR) at King Abdulaziz University, Jeddah: GPIP:1215-611-2024.

### Competing Interests

The authors declare that they have no competing interests.

### Author Contributions

- Ahmed Alzahrani conceived and designed the experiments, performed the experiments, performed the computation work, authored or reviewed drafts of the article, and approved the final draft.
- Muhammad Zubair Asghar performed the experiments, analyzed the data, prepared figures and/or tables, and approved the final draft.

### Ethics

The following information was supplied relating to ethical approvals (*i.e.*, approving body and any reference numbers):

The research protocol for this study was reviewed and approved by the Faculty of Computing, Gomal University Institutional Review Board (FOC-GU-IRB). The approval reference number is FOC-GU-IRB/2024-01. Ethical guidelines set by the institution were followed throughout the study, and all participants provided informed consent.

### Data Availability

The raw measurements are available in the Supplemental File.

## Supplemental Information

Supplemental information for this article can be found online at http://dx.doi.org/10.7717/peerj-cs.2698#supplemental-information.

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
