# Peer review of "Enhancing the prediction of vitamin D deficiency levels using an integrated approach of deep learning and evolutionary computing"

_PeerJ Computer Science, doi:10.7717/peerj-cs.2698_

## Round 0.1 · original submission · Major Revisions

Dear Authors,

Thank you for submitting your manuscript. Feedback from the reviewers is now available. It is not recommended that your article be published in its current format. However, we strongly recommend that you address the issues raised by the reviewers and resubmit your paper after making the necessary changes.

Best wishes,

Reviewer 1 ·

Basic reporting

A preliminary theoretical analysis of the proposed methodology motivates its rationale. However, this manuscript has many problems, such as structure, contribution, proposed method, etc., so it is unsuitable for publication.

Experimental design

A preliminary theoretical analysis of the proposed methodology motivates its rationale. However, this manuscript has many problems, such as structure, contribution, proposed method, etc., so it is unsuitable for publication.,

Validity of the findings

A preliminary theoretical analysis of the proposed methodology motivates its rationale. However, this manuscript has many problems, such as structure, contribution, proposed method, etc., so it is unsuitable for publication.,,

Additional comments

A preliminary theoretical analysis of the proposed methodology motivates its rationale. However, this manuscript has many problems, such as structure, contribution, proposed method, etc., so it is unsuitable for publication.

·

Basic reporting

Clarity and Language:
The manuscript is well-written and uses professional language throughout. However, some grammatical errors and minor typos (e.g., "eûective" instead of "effective") could benefit from a thorough proofreading.
The introduction provides a solid background on Vitamin D deficiency and highlights the motivation for the research effectively.

Context and Literature Review:
The literature review is adequate, referencing relevant works such as Sambasivam et al. (2020) and Patino-Alonso et al. (2022). However, the discussion on gaps in prior studies could be expanded with more concrete comparisons.
The research gap is clearly identified, emphasizing the novelty of integrating CNN, BiLSTM, and Genetic Algorithms (GA).

Structure:
The manuscript conforms to PeerJ standards, with well-defined sections such as Introduction, Materials & Methods, and Results.

Experimental design

Relevance and Scope:
The study fits well within the journal’s aims, offering an innovative AI-driven approach to healthcare prediction.

Methods and Replicability:
The methodology is robust, incorporating data preprocessing, model design (CNN+BiLSTM), and optimization (GA). The explanation of hyperparameter tuning is comprehensive.
The dataset description is clear, with ethical approval and informed consent explicitly mentioned. This ensures replicability.
Preprocessing steps, such as data balancing and label encoding, are appropriately detailed.

Evaluation Metrics:
The manuscript uses standard metrics (accuracy, precision, recall, F1-score) to evaluate model performance. These are appropriate for classification problems.
An ablation study to isolate the contributions of CNN, BiLSTM, and GA strengthens the findings.

Validity of the findings

Results and Analysis:
The results demonstrate significant improvements over baseline models (e.g., RF, SVM) with a 97% accuracy for the proposed CNN+BiLSTM+GA model. The performance enhancement is well-supported by experimental data.
Tables and figures are informative, though the captions could be expanded for standalone clarity.

Reproducibility:
The provision of code and datasets in the supplementary material ensures transparency and reproducibility.

Limitations and Future Work:
The manuscript acknowledges limitations, such as computational demands and dataset diversity. Suggestions for future work, like employing pre-trained models, are relevant.

Additional comments

Strengths:
Innovative integration of CNN, BiLSTM, and GA.
Thorough experimental setup and rigorous analysis.
Practical implications for non-invasive Vitamin D deficiency prediction in healthcare.

Areas for Improvement:
Strengthen the introduction with additional context on the impact of non-invasive healthcare predictions.
Expand the discussion of prior work to include more details on existing gaps.
Address computational efficiency in more depth, considering real-world deployment challenges.

·

Basic reporting

This study focuses on an important issue in health. The study is particularly significant due to the use of genetic algorithms to optimize the performance of deep neural networks. However, the authors did not provide a justification for using CNN and BiLSTM, especially when research has shown that models based on transformer architectures perform much better than CNN and BiLSTM models.

There is need to correct line 44, “Therefore, it is essential to develop more efficient non-invasive e predictive systems that can”

On line 73 the authors stated “Current studies on the prediction of VDD levels using ML and standard features have limitations [1,3].” But they didn’t state what are the current limitations in the status quo.
The research gap is not clearly defined or adequately justified. While the authors highlighted in the literature review that there is existing work on predicting vitamin D, they did not sufficiently explain why further research on this subject is necessary.

Additionally, the authors did not provide information on the F1 score, Recall, Precision, and Accuracy of the existing work, which would help clarify what they aim to improve with their approach. It would be highly beneficial for the authors to justify the need for using CNN, BiLSTM, and Genetic Algorithms (GA) in the context of the existing F1 score, Recall, Precision, and Accuracy metrics.

Experimental design

The authors did an excellent job of collecting data from scratch. This is a significant undertaking and makes the work highly contextual to their specific circumstances. The experimental design is generally sound and clear enough to be understood and reproduced. It is commendable that the authors employed an oversampling technique to balance their training data.

It is also recommended that the authors explore other encoding techniques (such as one-hot encoding or leave-one-out encoding) and evaluate their impact on the performance of the ML models. The use of convolutional layers to extract features built by the embedding layer is an interesting approach. However, it would be beneficial for the authors to provide readers with a diagrammatic representation of the model’s architecture for better understanding.

The statement in line 321 is somewhat confusing: “The Sigmoid technique was employed to determine the probability of each category: 'VDD1,' 'VDD2,' 'VDD3,' and 'VDD4.'” This appears inconsistent with the earlier claim that the activation function for the output layer is SoftMax.

Additionally, line 352 requires restructuring, as the content is not clearly visible.
It is recommended that the authors run an experiment comparing the use of Genetic Algorithms (GA) to optimize the DNN with conventional methods, such as grid search, to validate whether GA is indeed the most effective optimization method for this task.

There is further confusion in line 365. The authors mention using the "Mean Squared Error (MSE) metric for predicting vitamin D level"; however, given that the activation function for the output layer is SoftMax, this implies they were predicting a class, not a continuous value. In the dataset, there are two variables: “Vitamin D Level (ng/mL)” and “VDD Deficiency Level.” It is unclear which of these two variables was actually predicted. Clarification on this point would significantly improve the manuscript.

Validity of the findings

Table 3 does not clearly show how the iterations of the CNN+LSTM configurations resulted in the 10 different metrics. Additionally, it is unclear from Tables 2 and 4 how the results presented in Table 3 were derived. As previously mentioned, the metrics used in Table 3 indicate that this task was a classification task. However, the authors used MSE to optimize the CNN+LSTM model, which could lead to confusion. The authors are encouraged to maintain consistency throughout the paper to avoid misunderstandings.

It is noteworthy that the authors successfully ran five other models on the same dataset and compared the results with their model. It is particularly interesting that their method performed well compared to the existing models.

However, the general complexity of the algorithm is not clearly defined. Based on the provided function, it is unclear whether the complexity falls within the linear or polynomial range.

Additional comments

This is generally a good paper, however there are issues to be addressed as stated in comments above.

·

Basic reporting
* * *
Experimental design

--

Validity of the findings
* * *
Additional comments

1. Line 108: "into account the issue of multi co linearity" — There is a typo here; it should be "multicollinearity."
2. Line 117: "to deficiency in males (, with a significance level of p < 0.01)" — must be corrected like (p<0.01)
3. Line 120: "performance using different evaluation matrices" — The term "matrices" should be corrected to "metrics."
4. In Tables: Replace " F-score" with "F1-score"
5. Table 4: Correct "sizw" to "size."

AND MANY MORE. The manuscript should be thoroughly reviewed and revised to correct writing errors and improve the overall writing style.
6. Please provide detailed information about the feature selection process, including which method was used and which features were included in the model.
7. Multicollinearity problem must be checked. If it exists, the model's performance should be examined under the condition of multicollinearity.
8. The proposed model's robustness should be evaluated using various methods and testing scenarios.
9. Lines 41–42: "In the past, traditional techniques for predicting and diagnosing VDD involved testing the levels of serum 25-hydroxyvitamin D [15(OH)D] levels through biochemical assays" — Please correct "[15(OH)D]" to "25(OH)."
10. The structure of the manuscript is quite disorganized. The Results and Discussion sections should be restructured to enhance clarity and readability.
11. Carefully check the reference list; references 1–3 and 2–4 appear to be duplicates.
12. Please include a descriptive statistics table for all features used in the study. (Check the content of reference 2 in your reference list as a sample of the descriptive statistics table)
13. The conclusion must be rewritten to provide clear insights and actionable implications based on the study findings.
14. The title should be rewritten to reflect the methods used in the proposed model
15. Please paaphase this paragraph to prevent similarity:
C. Ablation study
471 The ablation study is carried out to assess the effectiveness of each part by eliminating the components and observing how the model works under various conditions. The following conclusions are reached from the dataset results (see Table 7):

---

## Round 0.2 · accepted · Accept

Dear Authors,

Reviewers have accepted your manuscript. Thank you addressing their concerns. Your manuscript now seems ready for publication.

Best wishes,

Reviewer 1 ·

Basic reporting

The authors have thoroughly addressed all my concerns, and I recommend that this paper be accepted for publication.

Experimental design

The authors have thoroughly addressed all my concerns, and I recommend that this paper be accepted for publication.

Validity of the findings

The authors have thoroughly addressed all my concerns, and I recommend that this paper be accepted for publication.

Additional comments

The authors have thoroughly addressed all my concerns, and I recommend that this paper be accepted for publication.

·

Basic reporting

The authors have cleared dealt with all my comments in their response.

Experimental design

The authors have cleared dealt with all my comments in their response.

Validity of the findings

The authors have cleared dealt with all my comments in their response.

Additional comments

I advise the editor to accept the new version of the document